# Observation of suppressed viscosity in the normal state of ³He due to superfluid fluctuations

Rakin N. Baten [1], Yefan Tian[1], Eric N. Smith[1], Erich J. Mueller [1] & Jeevak M. Parpia [1] ✉

Evidence of fluctuations in transport have long been predicted in ³He. They are expected to contribute only within $100\mu K$ of $T_c$ and play a vital role in the theoretical modeling of ordering; they encode details about the Fermi liquid parameters, pairing symmetry, and scattering phase shifts. It is expected that they will be of crucial importance for transport probes of the topologically nontrivial features of superfluid ³He under strong confinement. Here we characterize the temperature and pressure dependence of the fluctuation signature, by monitoring the quality factor of a quartz tuning fork oscillator. We have observed a fluctuation-driven reduction in the viscosity of bulk ³He, finding data collapse consistent with the predicted theoretical behavior.

The normal state of a superfluid contains transient ordered patches that grow as the system is cooled towards the transition temperature $T_c$. Observing the influence of these fluctuations on transport in liquid ³He has been a scientific goal that has been unfulfilled for nearly 50 years[1]. Similar fluctuations are found near other ordered states, such as magnets[2], superconductors[3], and alkali gases[4], where they are often related to pseudogap phemomena[5,6]. These fluctuations have been particularly well studied in ⁴He, where the extremely short coherence length allows the $\lambda$ anomaly in the heat capacity of ⁴He to serve as a model system for scaling[7]. Due to the low pairing energy and long coherence length, finding such signatures in ³He, however, has been challenging. Here we observe a fluctuation-induced suppression of the viscosity of bulk ³He near $T_c$. This provides crucial information about the transport signature which can be used to probe contemporary phenomena such as the topologically nontrivial nature of superfluidity in confined ³He[8].

The low-temperature normal state of ³He is our best example of a Fermi liquid, whose properties are understood in terms of a gas of interacting quasiparticles[9]. As the temperature is lowered, the phase space available for scattering is reduced and the mean time between scattering events grows as $\tau \propto T^{-2}$. As a consequence, transverse momentum gradients produce smaller stresses at low temperatures, quantified by the viscosity $\eta \propto \tau \propto T^{-2}$. A scattering resonance emerges as the liquid is cooled towards the superfluid transition, where particles form short-lived Cooper pairs during scattering events. Such resonances enhance the scattering, leading to a decrease in the viscosity. In a clean 3D system (such as ³He), this suppression occurs in only a very narrow window of temperature $\delta T = T - T_c$ where the pair lifetime $\tau_{GL} \approx \hbar/k_B\delta T$ is comparable to $\tau$. Thus one only expects to see a measurable reduction of the viscosity at temperatures of order 1% above $T_c$. In principle, the nature of these fluctuations will change when one is within the scaling regime[10,11] $\delta T/T_c = (T_c/T_F)^4 \approx 10^{-12}$, but in practice such precision is unachievable.

In addition to being of fundamental interest, the fluctuation contributions to transport are important for future experiments which will look for edge modes[12–14] in ³He as a signature of topological superfluidity[15–20]. The contribution to viscosity from these edge modes will be small, and accurate measurements will be needed to distinguish them from the effects of fluctuations. Here we report the necessary base-line measurements.

Fluctuation effects in ³He have previously been observed in the attenuation of zero (collisionless) sound[21–23], with ever-increasing experimental and theoretical sophistication[24–27]. While valuable, these are not a substitute for transport experiments. Observing the fluctuation contributions to viscosity is challenging and previous attempts[28–31] have had flaws that obscured or complicated the phenomena. In this work, we overcome these challenges.

Firstly, refs. 30,31 observed significant deviation from Fermi liquid behavior ($\eta \propto T^{-2}$) at all temperatures. Such deviations are unphysical, and are not seen in heat capacity[32], thermal conductivity[33], in

[1]Department of Physics, Cornell University, Ithaca, NY 14853, USA. ✉e-mail: jmp9@cornell.edu

collisonless sound measurements[21], or in previous measurements with quartz forks[34]. The deviations may be due to the temperature dependence of the properties of the metallic alloys used as vibrating elements[35]. We avoid this issue by using quartz forks.

Secondly, refs. 28,29 inferred temperature from the susceptability of a small sample of undiluted cerous magnesium nitrate (CMN). While accurate at ~10 mK, this approach suffers from systematic errors near the magnetic ordering temperature of CMN. Our current experiment uses a Lanthanum diluted CMN (LCMN) thermometer (Fig. 1a), referencing thermometry to the widely accepted PLTS2000 temperature scale[36,37].

Finally, our experiment takes pains to work within the hydrodynamic regime, where the viscous mean free path $\lambda_\eta$ is small compared to all other relevant length scales. In refs. 28,29, $\lambda_\eta$ was comparable to the cavity height at low pressure, leading to slip, and deviations from Fermi liquid behavior which obscured the influence of fluctuations. Torsional oscillator experiments[38] find that the contributions from these Knudsen effects become observable when the device dimensions are $d \approx 8\lambda_\eta$. In the present work, our fork has tines that are 0.61 mm wide × 0.253 mm thick × 3.64 mm long, spaced 0.194 mm apart, housed in a cylindrical casing ≈ 3 mm in diameter (Fig. 1b). The smallest of these dimensions, the 0.194 mm tine spacing, is more than 8 times $\lambda_\eta$ except at the very lowest temperatures (see Supplemental Note 1, Supplemental Table 1). Thus, Knudsen effects should be negligible.

## Results

We monitor the quality factor $Q = f_0/\Delta f$ of a quartz fork[34] immersed in liquid ³He cooled to mK temperatures by a nuclear demagnetization stage[39]. Here, $f_0$ is the resonant frequency and $\Delta f$ is the resonance linewidth. The oscillator damping can be related to the helium viscosity ($Q \propto \eta^{-1/2}$)[34], and we operate in the hydrodynamic regime. Temperature was measured with a diluted paramagnetic salt thermometer placed in the same ³He volume proximate to the quartz fork. Additional details on thermometry, fork operation, Fermi liquid viscosity, the hydrodynamic regime, and background subtraction are provided in the methods section and in Supplementary Notes 1 and 2. The pressure was maintained at a constant value using electronic feedback for each temperature sweep.

The data obtained at several pressures from 0.5 bar to 29.3 bar are shown in Fig. 2a. For each data set, we show the best linear fit as a dashed line passing through the origin, corresponding to the Fermi liquid prediction $\eta \propto T^{-2}$ (i.e., $Q \propto T$). In Fig. 2b, we compare the value of $Q/T$ obtained at all pressures near $T_c$, illustrating the extent of the departure from Fermi liquid behavior near $T_c$.

As $T_c$ is approached from above (Fig. 3a), a small increase in $Q$ ($\delta Q$) is observed relative to the dashed line, corresponding to a suppression of $\eta$. At high pressure, the deviations are large enough that $Q$ actually passes through a minimum in the normal state. At low pressure, $\delta Q$ is

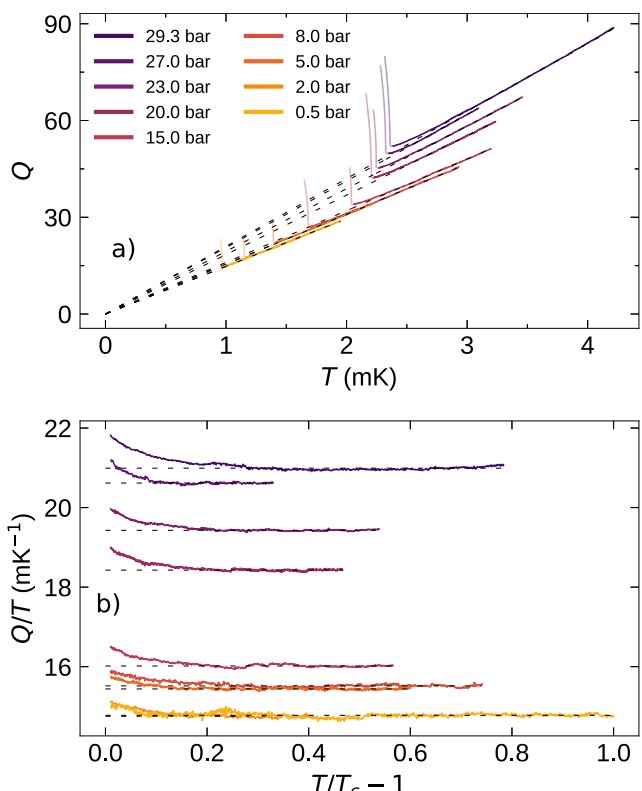

**Fig. 2 | Quartz fork $Q$ vs temperature. a** The inferred $Q$ of the quartz fork at various pressures *vs* the temperature. The expected Fermi liquid behavior was obtained after processing described in Supplementary Note 2, ($Q \propto \eta^{-1/2} \propto T$), and is represented by dashed lines. The superfluid transition is marked by an abrupt increase in the $Q$, and data below $T_c$ is shown as lighter-shaded lines. **b** Plot of $Q/T$ vs $(1 - T/T_c)$. This plot illustrates the extent of the departure of $Q$ from linear behavior with pressure.

smaller, though it can be resolved. The differences between high and low-pressure results are highlighted in Fig. 3b and its insets. Upon entering into the superfluid state the $Q$ sharply increases due to the rapid decrease in viscosity[28,40–42] at $T_c$. The quality of the data is sufficient to illustrate the development of $\delta Q$ in Fig. 3c with pressure.

## Discussion

Proximity to superfluidity enhances quasiparticle scattering: Quasiparticles that pass near each other form short-lived pairs, increasing the scattering rate, $1/\tau$. The viscosity is proportional to the scattering time $\tau$, ($\propto T^{-2}$), which is therefore suppressed near $T_c$. Emery[1] writes the fluctuation contribution to the viscous scattering time $\tau$ as

$$\frac{\delta\tau}{\tau} = -\Gamma\left(\frac{k_B T_F \tau}{\hbar}\right)(k_F \xi_{00})^{-3}\alpha\left(1 - \frac{\theta^{1/2}}{\alpha}\tan^{-1}\frac{\alpha}{\theta^{1/2}}\right) \quad (1)$$

where the quantity $\delta\tau$ is the additional scattering time due to the broken pairs above $T_c$, and $\alpha$ is a fitting constant. Here $\theta = \frac{T}{T_c} - 1$ is the reduced temperature, $T_F$ is the Fermi temperature, and $\Gamma$ is a numerical constant that depends on the pairing and the transport parameter (in this case viscosity, $\eta$). The unitless quantity $k_F \xi_{00}$ is the product of the Fermi wavevector and the pairing coherence length, and in bulk ³He can be expressed as

$$(k_F \xi_{00})^2 = \frac{7\zeta(3)}{12\pi^2}\left(\frac{T_F}{T_c}\right)^2 \quad (2)$$

where $\zeta(3) \approx 1.2$ is Apéry's constant and $\zeta$ is the Riemann Zeta function.

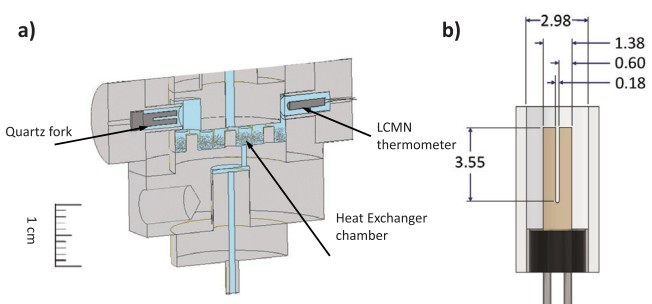

**Fig. 1 | Cell Schematic. a** The location of the quartz fork and LCMN thermometer are shown in relation to the heat exchanger. **b** Schematic image of the quartz fork with dimensions in millimeters.

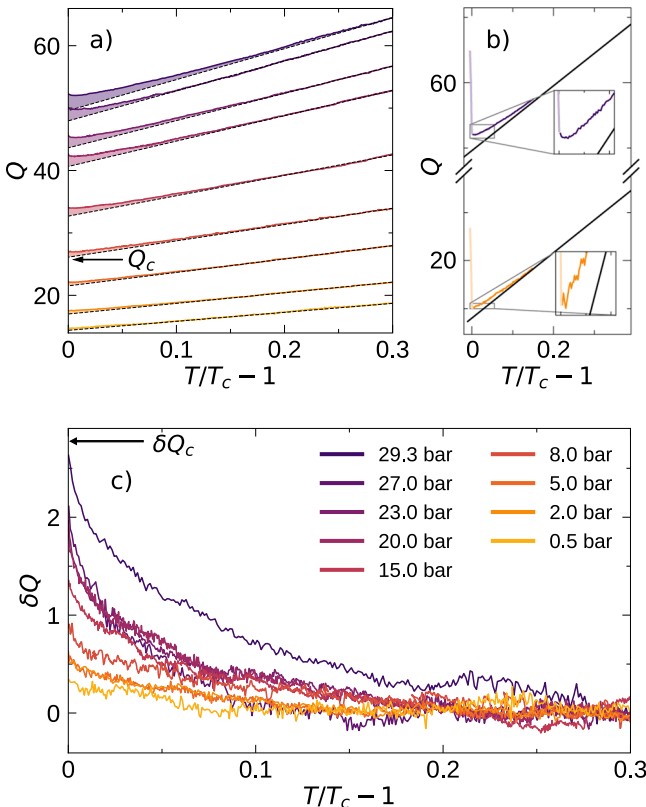

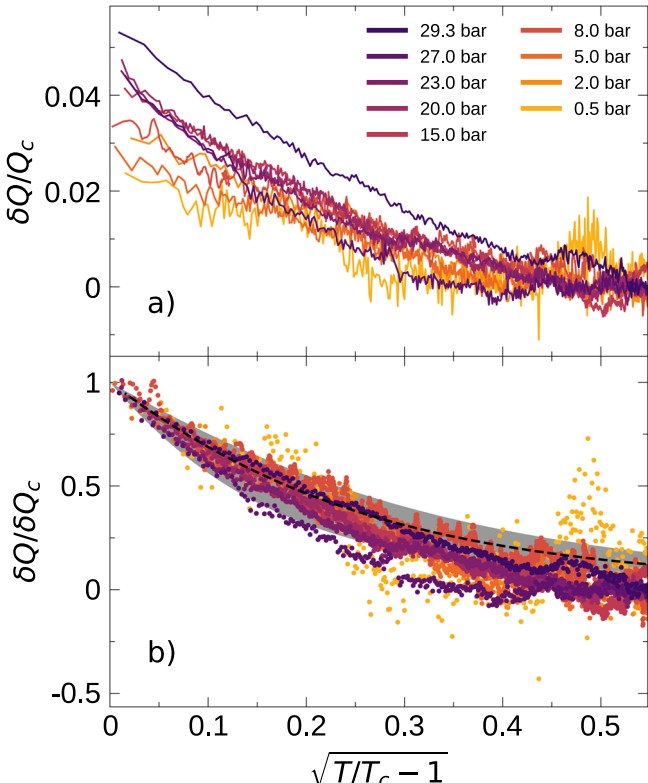

**Fig. 3 | Quartz fork $Q$ vs temperature near $T_c$. a** Departure from the Fermi liquid behavior (linear slope) is seen at all pressures just above $T_c$. Also marked is the value of $Q_c$ for the 8-bar run. **b** $Q$ vs $(T/T_c − 1)$ near $T_c$ of the 29.3 bar and 2 bar runs. It is evident from the insets that the higher pressure run shows a minimum in the $Q$ before $T_c$ is attained, while the lower pressure data shows no minimum. **c** The excess $Q$ vs $T/T_c − 1$.

**Fig. 4 | Normalized fluctuation contribution vs reduced temperature. a** The measured excess $Q$ (see Fig. 3) of the quartz fork at various pressures, normalized to $Q_c$ (see Fig. 3c) plotted against the square root of reduced temperature. This plot shows that the contribution to $Q$ of the fluctuation component increases faster than the increase of $Q_c$ with pressure. **b** $\delta Q$ normalized to $\delta Q_c$ (see Eqs. (3), (4)). The dashed line shows the expected temperature-dependent fit to the fluctuation component of viscosity in Eq. (1) (see ref. 1). The shaded gray region represents the $1\sigma$ range in the curve fit based upon the error in the fit parameters.

Since the $Q \propto \eta^{-1/2} \propto \tau^{-1/2}$, it follows that $\delta Q/Q = -1/2\delta\tau/\tau$. We can rewrite $\tau(T) = \tau_c \times (T_c/T)^2$ and $Q(T) = Q_c \times T/T_c$. $Q_c$ is the value of the $Q$ at $T_c$ without the contribution due to fluctuations (See Fig. 3b). Thus $(\delta\tau/\tau) = -2 \, (\delta Q/Q_c) \times (T_c/T)$. This yields a modified version of Equation (1),

$$\frac{\delta Q(T)}{Q_c} = C(P)\frac{\alpha}{1+\theta}\left(1 - \frac{\theta^{1/2}}{\alpha}\tan^{-1}\frac{\alpha}{\theta^{1/2}}\right), \qquad (3)$$

and

$$C(P) = \frac{1}{2}\Gamma\left(\frac{k_B T_F \tau_c}{\hbar}\right)(k_F\xi_{00})^{-3}. \qquad (4)$$

We can extract $Q_c$ from the linear fits in Fig. 3 and plot the ratio $\delta Q/Q_c$ from Eq. (3) in Fig. 4a. For small $\theta$, Eq. (3) has the form $\delta Q \approx \delta Q_c(1 - \pi\theta^{1/2}/2\alpha)$, where $\delta Q_c$ is the excess $Q$ at $T_c$. Thus, it is natural to use $\theta^{1/2}$ as the horizontal axis. Both $Q_c$ and $\delta Q$ increase with pressure, but $\delta Q$ has a slightly stronger dependence: The ratio $\delta Q_c/Q_c$ varies from ~2% at the lowest pressure measured to ~5% at the highest. The corresponding values of the zero sound attenuation coefficient, A, $\delta$A/$A_c$ measured in collisionless sound varied from ~8% at 32.56 bar, ~6.5% at 19.94 bar, and "very approximately 2%" at 0.05 bar[21]. Assuming that $\alpha$ is not pressure dependent, Eq. (3) predicts that the excess $Q$'s should collapse if normalized as $\delta Q/\delta Q_c$. In Fig. 4b, we test that feature, showing Emery's prediction as a black dashed line, using $\alpha = 0.43$. The agreement is quite remarkable, with slight deviations at larger values of $\theta^{1/2}$.

We further quantify this agreement by independently fitting each fixed-pressure run to Eq. (3), extracting our best estimates of

the pressure dependence of $\delta Q_c = C(P)\alpha Q_c$ and $\alpha$. As seen in Fig. 5a, any pressure dependence of $\alpha$ is weak. The contributions to $C(P)$ in Eq. (4) are reasonably well known. We take $\eta T^2$ from ref. 43 to calculate $\tau_c$ (after correction for temperature scales), and $v_F$, n, and $m^*/m$ from refs. 32,44; (See Supplementary Note 1 for more details). Emery argues that $19.5 < \Gamma < 46.8$ for $p$ wave pairing, with the true value likely lying in the middle of that range. We treat $\Gamma$ as a free parameter, finding a best-fit value $\Gamma = 40.8$, which is at the upper end of the expected range. Nonetheless, the resulting curve, shown in Fig. 5b, agrees very well with our measurements. The error bars on $\alpha$, $\delta Q_c/Q$ in Fig. 5a, b represent a $1\sigma$ standard deviation. The error bars on $\alpha$ are derived from the calculation of the fit to Equation (3) and random noise error in $Q$; the error bars on $\delta Q_c/Q_c$ in Fig. 5b are derived from the error in $\delta Q_c$ (the error in $Q_c$ is negligible in comparison to $\delta Q_c$).

The somewhat large value of $\Gamma$ may be the result of limitations in Emery's modeling. Lin and Sauls[27] argued that Emery's calculation contains some double-counting, and that it incorrectly included interference terms among the different scattering channels. Another source of theoretical uncertainty is the scattering time $\tau$ which we used in evaluating Eq. (4). In any event, the magnitudes of the fluctuation contribution to the viscosity are seen to be smaller than the values noted in refs. 21,22.

With improvements in signal recovery using low-temperature amplifiers, the precision and noise of the excess $Q$ could be greatly improved, and perhaps used to measure the pressure dependence of the Landau parameter $F_2^s$ as was proposed for collisionless sound[27]. The values of $F_2^s$ are poorly known[27], as they are derived from the

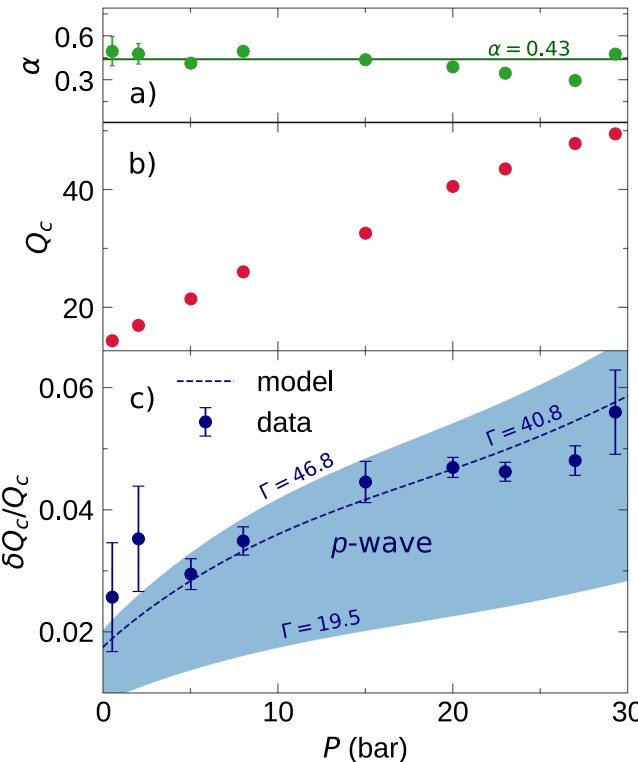

**Fig. 5 | Comparison of measured and calculated parameters. a** The values of $\alpha$ obtained to yield the fit shown in Fig. 4. **b** The horizontal line denotes the mean $\alpha$. **b** Values of $Q_c$ based on the linear fits for Fermi liquid behavior shown in Fig. 3. **c** $\delta Q_c$, normalized by $Q_c$ plotted against the pressure. The dashed line shows the expected temperature-dependent fit to the normalized fluctuation component in Eq. (3) (see ref. 1) based on previously measured values of $\eta T^2$ (ref. 43), $v_F$, $n$, and $m^*/m$ (refs. 32,44). Details in Supplementary Note 1, Supplementary Table 1. Shaded blue region marks the variation of $\Gamma$ for $p$ wave pairing in the model by Emery (ref. 1).

pressure dependence of the attenuation of transverse zero sound which a difficult-to-measure parameter[45].

Looking forward, an important next step will be to extend these measurements to strongly confined geometries, where topological surface states appear[12–20]. In such geometries $T_c$ can be significantly suppressed[18], leaving an extended region where fluctuations can potentially become stronger. Experiments studying thermal transport in such narrow channels[19] reveal a crossover between bulk and surface-dominated regimes, which depend on surface quality[18,46,47]. The role of pairing fluctuations, and their interaction with surface modes, has not yet been established, and will be the focus of future research. For the present study conducted in bulk $^3$He, the impact of surface states (that exist only below $T_c$) on fluctuations should be negligible.

We have observed that incipient pairing fluctuations contribute a small but significant portion of the scattering above $T_c$. This contribution is resolved at all pressures, and is comparable to that observed using the attenuation of collisionless (zero) sound. There are significant efforts underway to study transport processes such as mass and spin edge currents[12,13,48], thermal Hall effects[14], thermal conductivity[19], and spin diffusion in highly confined geometries, where the suppression of $T_c$ and strong confinement should lead to the enhancement of the contribution of fluctuations, potentially impacting exotic topological transport.

## Methods

### Quartz fork

The experimental results described here were obtained with a quartz fork[34] with dimensions much greater than the quasiparticle mean free path. The other relevant length scale is the viscous penetration depth, $\delta = (2\eta/\rho\omega)^{1/2}$, where $\eta$ and $\rho$ are the viscosity and density of the $^3$He, while $\omega$ is the resonant frequency of the fork. The largest value of the viscous penetration depth occurs at $T_c$ at 0 bar. Unlike collisionless sound where $\omega\tau \geq 1$, here the fork operates in the hydrodynamic limit ($\omega\tau \leq 1$) with $\omega = 2\pi f_0 \approx 2 \times 10^5\,\mathrm{s}^{-1}$ and $\tau \approx 2 \times 10^{-6}\,s$ at $p = 0$ bar and $T = T_c$ (see Supplementary Note 1 for further details).

### Fork operation

The quartz fork was operated in a phase-locked loop and driven at a fixed drive voltage. The phase-locked loop was set to drive the fork at a frequency fixed to within 5 Hz from resonance. When the frequency shift exceeded these bounds, the drive frequency was adjusted to bring the device on resonance again. The resonant frequency and $Q$ were inferred from the complex response recorded by the lock-in amplifier. In order to simplify this conversion, a significant background response of the non-resonant signal ("feedthrough") had to be measured and subtracted from the received signal. After subtraction, when the drive frequency was swept through resonance, the signal was seen to be Lorentzian, and was calibrated to yield the $Q$. Further details are provided in Supplementary Note 2.

### Thermometry

Thermometry was accomplished using a small pill (1.25-mm diameter, 1.25 mm high) of ≤30 μm diameter powdered LCMN, packed to 50% density. The pill and monitoring coil were located in a niobium shielding can. The coil structure consisted of an astatically wound secondary and primary coil. The primary coil was driven at constant voltage through a 10 kΩ resistor by a signal generator at a fixed frequency (23 Hz). The secondary coil was coupled to the input of a SQUID. The secondary loop had an additional mutual inductor to allow the cancellation of the induced signal in the loop. The input of this mutual inductor was driven by the same signal generator as the primary. The drive amplitude and phase of this cancellation signal were stepped by discrete amounts to cancel out most of the current in the secondary loop. The drive applied to the mutual inductor and the magnitude of the received signal was proportional to the susceptibility of the LCMN. These were calibrated against a melting curve thermometer and against the superfluid transition temperatures at various pressures. The thermometer had a resolution of better than 50 nK.

## Data availability

The data generated in this study and shown in all the plots in this paper and the supplementary material have been deposited in the Cornell University e-commons data repository database under accession code https://doi.org/10.7298/r4jy-py94.

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

## Acknowledgements

This work was supported by the National Science Foundation, under DMR-2002692 (J.M.P.), and PHY-2110250 (E.J.M.).

## Author contributions

Experimental work was principally carried out by Y.T. and R.B. with further support from E.N.S. and J.M.P. Analysis and the presentation of figures were carried out by R.B. and Y.T. We thank Anna Eyal for generating the figure of the cell in Fig. 1. E.M. significantly contributed to the analysis and the writing of the manuscript, J.M.P. supervised the work and J.M.P., and E.M. had leading roles in formulating the research and writing this paper. R.B. and Y.T. contributed equally to the publication of this result. All authors contributed to revisions to the paper.

## Competing interests

The authors declare no competing interests.
