## [Peer Review File · Nature Communications]

Reviewer #1 (Remarks to the Author):

The manuscript by Baten et al describes measurements of the damping a quartz tuning fork immersed in liquid ^3He as a function of pressure and temperature above the superfluid transition temperature T_c . Anomaly of damping is observed in the vicinity of T_c , which is attributed to suppressed viscosity due to superfluid fluctuations. While qualitatively the effect was observed earlier, the authors claim to provide the first precise quantitative measurement, which can be used in future to understand transport properties of superfluid ^3He in complicated geometries.

Unfortunately, the claim of the precise measurements is shadowed by the deficiency in the measurement procedure related to accounting for the electrical background in the fork measurements. To correct for this deficiency, the authors employ complicated procedure, which essentially uses the assumption of ideal Fermi-liquid behavior of ^3He at temperatures further from T_c . I find this approach not very satisfactory. One would think that by using full spectra sweeps and assuming Lorentzian response of the fork and reasonable background behavior, which is suggested by supplementary figures 3-5, it should be possible to figure out damping correction without attributing particular physical properties to the studied system. Or background can be determined experimentally by repeating exactly the same measurements, but with a slightly different pressure, so that the fork resonance is moved away from the frequencies in question.

Besides needed improvements in the methodology, the manuscript also lacks an example of application of the collected knowledge to uncover new physics of the topological superfluid. Without such component of wide interest, the work might suit better a more specialized journal. Thus I cannot recommend publication in Nature Communications in the present form.

Below are some additional comments.

The phrase "remarkable collapse" in the abstract is misleading. I originally thought that this is exactly where new physics appears, that is the fluctuations are suddenly suppressed where they should exist according to current knowledge. In reality it seems to refer to the collapse of the measured data on each other when scaled according to existing models.

The first two sentences of introduction are also misleading. The first sentence speaks about superfluids in general, while the second switches to ^3He without specifying that. In superfluid ^4He fluctuations are actually well known.

References 19 and 20 are the same.

α is not defined at the first appearance (Eq. 1).

Exact fork geometry is not specified.

On line 6 of page 7 of supplementary materials, ω is probably missing.

In supplementary figure 8 caption, "dashed red line" seems to be orange and is called in the main text like this.

Reviewer #2 (Remarks to the Author):

The manuscript reveals the vital role of superfluid fluctuations in bulk ^3He in the normal state near the superfluid transition temperature T_c . By monitoring the quality factor of a quartz fork immersed in liquid ^3He , the authors observed suppression of viscosity from the Fermi liquid behavior without fluctuations. In the manuscript, they also discuss the deviation of the quality factor compared with Emery's theoretical prediction. The experimental results are in good agreement with Emery's predictions.

The main impact of the paper is the observation of the effect of incipient pairing fluctuations through the measurement of the transport coefficient. The manuscript reads well and is well understandable for a reader working in the broad area of the quantum fluid. In terms of substance, I recommend its publication in Nature Communications after clarifying the following minor points.

1. As mentioned on page 3 of the main manuscript, the previous experiments in Refs. 28 and 29 observed the suppression of viscosity near T_c (e.g., Fig.1 in Ref. 28). Does the viscosity reduction in previous studies essentially differ from those observed in the present experiment? In other words, can the authors extract the consequences of the Knudsen regime by comparing the previous and present experimental results? I would like to see more arguments on this issue if possible.

2. On page 4, the authors briefly mention that the viscous mean free path is smaller than the quartz fork or chamber dimensions. It would be helpful to describe the details of the experimental setup (e.g., the size of the chamber) and the order of the viscous mean free path.

Reviewer #3 (Remarks to the Author):

The present article describes precision measurements in which a mechanical resonator is used to record fluctuation-based changes in the normal Fermi fluid ^3He in the vicinity of the superfluid transition. These are very challenging experiments due to the required extreme temperature, pressure and readout stability, conducted to an impressive standard. As an example of the level of challenge in measuring fluctuations in this system, the shot noise of mechanical collisions with broken Cooper pairs (normal component) on the other side of the superfluid transition, or anywhere below it, has never been observed in experiments. This is much owing to the extremely low characteristic energies in the system. The offered theoretical description of the data in the present manuscript is sound and the presentation generally clear. The general context/outlook of this paper is that, owing to recent development of nanotechnology, the microstructure and micromanipulation of superfluid ^3He has become within direct experimental reach, and understanding fluctuations is a central requirement for the exploration of this regime. This, in principle, easily justifies publication in Nature Communications: Superfluid ^3He is the only triplet (p wave) macroscopic quantum system where such experiments can be carried out. However, several aspects need clarification before the paper can be considered for publication. I have detailed my main observations below.

Recent work by the same authors [Nat Commun 11, 4843 (2020)] and by other groups, for example Ref 18 in the manuscript and [Nat Commun 11, 4742 (2020)], suggest that the thermal properties of superfluid ^3He within a few coherence lengths from a wall are rather different from those in the bulk superfluid. For example, heat can possibly be carried along surfaces without the bulk being involved. This property arises because the bulk superfluid is suppressed near walls. Could the authors comment on the effect of walls on the fluctuations measured here near the surface of the probe fork?

On page 3 the authors write “However, these experiments were carried out in a parameter space where the viscous mean free path was of order the confinement size ($\approx 95 \mu\text{m}$). Thus, the experiments were conducted in the slip dominated “Knudsen” regime which led to a modification of the effective viscosity away from the usual Fermi liquid behavior.” Could the authors explain

precisely where in the parameter space is the present experiment located, thus avoiding these difficulties? That is, what is different in practice?

In the concluding paragraph the authors write "This contribution is resolved at all pressures and is comparable to that observed using the attenuation of collisionless (zero) sound." It is not clear how the authors arrived at this conclusion. Could the authors either expand this discussion or explicitly point to the part of the manuscript where this is discussed?

The concluding paragraph points out that fluctuations of the sort measured in the bulk in this article are likely to become stronger in confined geometries. It would be useful to explain the expected/speculated consequences here, especially as these consequences are of crucial importance according to the abstract but not discussed anywhere in the main text.

The background subtraction analysis is shown in the supplementary material (supplementary figure 3) for a resonance curve where the Q value is by approximation around $31500/400=78$. This is bigger than for any of the data shown in Fig 1 that all the analysis that follows is based on. Thus, the background subtraction is easier in the example shown than in the actual analysis. Could this be replaced or accompanied by an example that is directly relevant for the data analysis? How does it work for $Q < 15$ (resonance width $> 2000\text{Hz}$) where the low-pressure data is extracted?

Minor technical and editorial notes:

The authors could use the Latex package Slunitx to produce roman " μ " in units such as " μm " and " μK "

The quantities α and Q_c are not defined where they first appear in the main text (page 6), and Q_c is only defined in a figure caption.

On page 4 the authors introduce the notation δQ to denote a small increase in Q . It would make the text easier to read if they stuck to this notation after that point instead of repeating the definition every time this quantity appears in the text in that Section and the related figures.

In figure 4 caption, the statement about the supplementary note could be c

References 19 and 20 point to the same article

In Methods, the fork operation is described as "... set to drive the fork at a fixed frequency ± 5 Hz from resonance." I believe the authors mean "... set to drive the fork at a frequency fixed to within 5 Hz from resonance."

We thank the reviewers for the opportunity to revise our paper in response to all their comments. We include our replies to the reviewers (in bold) below, and you will find the modifications to the text in red in the pdf supplied accompanying this reply.

A brief list of all the major modifications follows. All page numbers refer to the version of the ms where we show track changes (in red):

1. Addition of Figure 1, to provide a view of the experimental set up and fork dimensions.
2. Addition of Figure 2 that includes an expanded version of the old Fig 1a) and the addition of Fig 2 b) showing Q/T vs T/T_c-1 .
3. Slightly modified last sentence in the Abstract.
4. Addition of the statement on the scaling region associated with the λ point in ^4He in the introductory paragraph.
5. Substantial changes to the discussion on the mean free path in the last paragraphs of the introduction (p4).
6. Modified the colors in figures associated with various pressures to make them more friendly for persons with color blindness.
7. Modified the introductory section (see portions in red on p3, p4) to address concern about departure from Fermi liquid behavior.
8. Added details from reference [21] concerning the magnitude of the departure from Fermi-liquid behavior in ultrasound measurements [p10].
9. Modified original Figure 3 (new figure 5) to improve clarity.
10. Added a paragraph concerning surface states at the end of the discussion section (p14).
11. Added several new references.
12. Added rows to Supplementary Table 1.
13. Added 3 figures (S14-16) to Supplementary Note 2 concerned with low Q sweeps.

REVIEWER COMMENTS

Reviewer #1 (Remarks to the Author):

The manuscript by Baten et al describes measurements of the damping a quartz tuning fork immersed in liquid ^3He as a function of pressure and temperature above the superfluid transition temperature T_c . Anomaly of damping is observed in the vicinity of T_c , which is attributed to suppressed viscosity due to superfluid fluctuations. While qualitatively the effect was observed earlier, the authors claim to provide the first precise quantitative measurement, which can be used in future to understand transport properties of superfluid ^3He in complicated geometries.

Unfortunately, the claim of the precise measurements is shadowed by the deficiency in the measurement procedure related to accounting for the electrical background in the fork measurements. To correct for this deficiency, the authors employ complicated procedure, which essentially uses the assumption of ideal Fermi-liquid behavior of ^3He at temperatures further from T_c . I find this approach not very satisfactory. One would think that by using full spectra sweeps and assuming Lorentzian response of the fork and reasonable background behavior, which is suggested by supplementary figures

3-5, it should be possible to figure out damping correction without attributing particular physical properties to the studied system. Or background can be determined experimentally by repeating exactly the same measurements, but with a slightly different pressure, so that the fork resonance is moved away from the frequencies in question.

We thank the reviewer for his comments. We have expanded our discussion of previous experiments (pages 3, 4). While two experiments [30, 31] observed non-Fermi liquid behavior in viscosity away from the fluctuation regime, there is no evidence for such non-Fermi liquid behavior in experiments using torsion oscillators [28, 37], or in heat capacity [32], thermal conductivity [33] or in zero sound [21]. We note in the text that there is no theoretical basis for this behavior, which extends over a very large temperature regime, and we speculate that it is an artifact of the temperature dependent properties of the alloys used in their apparatus.

While our data analysis requires many steps, we believe that each of them is well justified. We would find similar results if we used a procedure which did not use the constraints of Fermi Liquid theory, but do not see the harm of using well established physical principles in our analysis. The reviewer makes two suggestions for alternative data analysis techniques. We do indeed use full spectral sweeps as part of our calibration procedure – but that technique is much slower than the one we use for the bulk of our data acquisition and would preclude the fine thermal resolution which we are able to produce. The second suggestion – comparing sweeps with slightly different pressures -- has challenges associated with it, and we do not believe that it would work without extensive modifications. There are only small frequency shifts associated with small pressure changes. Moreover, our background appears to be time-dependent, with a timescale of order days.

Besides needed improvements in the methodology, the manuscript also lacks an example of application of the collected knowledge to uncover new physics of the topological superfluid. Without such component of wide interest, the work might suit better a more specialized journal. Thus I cannot recommend publication in Nature Communications in the present form.

We have extended our discussion (last para of discussion section, p14) of how one needs a quantitative handle on fluctuations in order to explore the role of edge modes. This is particularly important because the narrow channels that will be used to look for edge modes will have an extended window of fluctuations due to the suppressed T_c from confinement.

Below are some additional comments.

The phrase "remarkable collapse" in the abstract is misleading. I originally thought that this is exactly where new physics appears, that is the fluctuations are suddenly suppressed where they should exist according to current knowledge. In reality it seems to refer to the collapse of the measured data on each other when scaled according to existing models.

We have modified the concluding sentence of the abstract

The first two sentences of introduction are also misleading. The first sentence speaks about superfluids

in general, while the second switches to ^3He without specifying that. In superfluid ^4He fluctuations are actually well known.

We have modified the introductory paragraph accordingly.

References 19 and 20 are the same.

This has been corrected

α is not defined at the first appearance (Eq. 1).

This has been corrected

Exact fork geometry is not specified.

We have added a figure (Figure 1) and also provide dimensions on Page 4, 5

On line 6 of page 7 of supplementary materials, ω is probably missing.

This has been corrected

In supplementary figure 8 caption, "dashed red line" seems to be orange and is called in the main text like this.

This has been corrected

Reviewer #2 (Remarks to the Author):

The manuscript reveals the vital role of superfluid fluctuations in bulk ^3He in the normal state near the superfluid transition temperature T_c . By monitoring the quality factor of a quartz fork immersed in liquid ^3He , the authors observed suppression of viscosity from the Fermi liquid behavior without fluctuations. In the manuscript, they also discuss the deviation of the quality factor compared with Emery's theoretical prediction. The experimental results are in good agreement with Emery's predictions.

The main impact of the paper is the observation of the effect of incipient pairing fluctuations through the measurement of the transport coefficient. The manuscript reads well and is well understandable for a reader working in the broad area of the quantum fluid. In terms of substance, I recommend its publication in Nature Communications after clarifying the following minor points.

1. As mentioned on page 3 of the main manuscript, the previous experiments in Refs. 28 and 29 observed the suppression of viscosity near T_c (e.g., Fig.1 in Ref. 28). Does the viscosity reduction in previous studies essentially differ from those observed in the present experiment? In other words, can the authors extract the consequences of the Knudsen regime by comparing the previous and present experimental results? I would like to see more arguments on this issue if possible.

We have included a new figure (Fig 1) that explicitly gives the dimensions of the fork and include a discussion in the concluding paragraph of the introductory section. We also provide new lines at the bottom of supplementary table 1 in supplementary note 1 that provides numerical values for the various lengths. Due to thermometric issues we are unable to compare the results from Ref 30 and 31 to those in the present paper. There are further complexities having to do with stiffening of the torsion rod from the ^3He inside it [29].

2. On page 4, the authors briefly mention that the viscous mean free path is smaller than the quartz fork or chamber dimensions. It would be helpful to describe the details of the experimental setup (e.g., the size of the chamber) and the order of the viscous mean free path.

We have included a new figure (Fig 1) that explicitly gives the dimensions of the fork and include a discussion in the concluding paragraph of the introductory section (p4, p5). We also provide new lines at the bottom of supplementary table 1 in supplementary note 1 that provides numerics of various length scales including a comparison of the mean free path to the fork dimensions.

Reviewer #3 (Remarks to the Author):

The present article describes precision measurements in which a mechanical resonator is used to record fluctuation-based changes in the normal Fermi fluid ^3He in the vicinity of the superfluid transition. These are very challenging experiments due to the required extreme temperature, pressure and readout stability, conducted to an impressive standard. As an example of the level of challenge in measuring fluctuations in this system, the shot noise of mechanical collisions with broken Cooper pairs (normal component) on the other side of the superfluid transition, or anywhere below it, has never been observed in experiments. This is much owing to the extremely low characteristic energies in the system. The offered theoretical description of the data in the present manuscript is sound and the presentation generally clear. The general context/outlook of this paper is that, owing to recent development of nanotechnology, the microstructure and micromanipulation of superfluid ^3He has become within direct experimental reach, and understanding fluctuations is a central requirement for the exploration of this regime. This, in principle, easily justifies publication in Nature Communications: Superfluid ^3He is the only triplet (p wave) macroscopic quantum system where such experiments can be carried out. However, several aspects need clarification before the paper can be considered for publication. I have detailed my main observations below.

Recent work by the same authors [Nat Commun 11, 4843 (2020)] and by other groups, for example Ref 18 in the manuscript and [Nat Commun 11, 4742 (2020)], suggest that the thermal properties of superfluid ^3He within a few coherence lengths from a wall are rather different from those in the bulk superfluid. For example, heat can possibly be carried along surfaces without the bulk being involved. This property arises because the bulk superfluid is suppressed near walls. Could the authors comment on the effect of walls on the fluctuations measured here near the surface of the probe fork?

Since these fluctuations occur because of incipient pairing above T_c and bound states occur below T_c we do not believe that bound states should influence the measurements done here above T_c . We do

include a short paragraph addressing effects due to strong confinement (p3) and also on p14, p15.

On page 3 the authors write "However, these experiments were carried out in a parameter space where the viscous mean free path was of order the confinement size ($\approx 95 \mu\text{m}$). Thus, the experiments were conducted in the slip dominated "Knudsen" regime which led to a modification of the effective viscosity away from the usual Fermi liquid behavior." Could the authors explain precisely where in the parameter space is the present experiment located, thus avoiding these difficulties? That is, what is different in practice?

We have included a new figure (Fig 1) that explicitly gives the dimensions of the fork and include a discussion in the concluding paragraph of the introductory section. We also provide new lines at the bottom of supplementary table 1 in supplementary note 1 that provides numerical values for various length scales. We include a note that the onset of departure from $\eta T^2 = \text{constant}$ behavior is observed for $d/\lambda = 8$ in Ref[37], and that this value is only attained (in our experiment) at low pressure where the fluctuation effects are minimal. Thus, Knudsen effects contribute only minimally to the departures from Fermi liquid behavior seen here.

In the concluding paragraph the authors write "This contribution is resolved at all pressures and is comparable to that observed using the attenuation of collisionless (zero) sound." It is not clear how the authors arrived at this conclusion. Could the authors either expand this discussion or explicitly point to the part of the manuscript where this is discussed?

We have added a sentence to clarify this statement in the bottom paragraph of page 10 quoting the values from Paulson and Wheatley's PRL Ref [21].

The concluding paragraph points out that fluctuations of the sort measured in the bulk in this article are likely to become stronger in confined geometries. It would be useful to explain the expected/speculated consequences here, especially as these consequences are of crucial importance according to the abstract but not discussed anywhere in the main text.

We have added a paragraph ahead of the conclusion (page 14 in the paper) that hopefully addresses this.

The background subtraction analysis is shown in the supplementary material (supplementary figure 3) for a resonance curve where the Q value is by approximation around $31500/400=78$. This is bigger than for any of the data shown in Fig 1 that all the analysis that follows is based on. Thus, the background subtraction is easier in the example shown than in the actual analysis. Could this be replaced or accompanied by an example that is directly relevant for the data analysis? How does it work for $Q < 15$ (resonance width $> 2000\text{Hz}$) where the low-pressure data is extracted?

We have added a brief analysis of a low pressure low Q run in the supplemental information (supp figs 14, 15, 16).

Minor technical and editorial notes:

The authors could use the Latex package Slunitx to produce roman “ μ ” in units such as “ μm ” and “ μK ”

The quantities α and Q_c are not defined where they first appear in the main text (page 6), and Q_c is only defined in a figure caption.

This has been corrected

On page 4 the authors introduce the notation δQ to denote a small increase in Q . It would make the text easier to read if they stuck to this notation after that point instead of repeating the definition every time this quantity appears in the text in that Section and the related figures.

This has been corrected

In figure 4 caption, the statement about the supplementary note could be c

This has been corrected

References 19 and 20 point to the same article

This has been corrected

In Methods, the fork operation is described as "... set to drive the fork at a fixed frequency ± 5 Hz from resonance." I believe the authors mean "... set to drive the fork at a frequency fixed to within 5 Hz from resonance."

This has been corrected

REVIEWERS' COMMENTS

Reviewer #1 (Remarks to the Author):

The authors took into account all technical suggestions and also improved discussion of physics, to demonstrate clearer the difference of this work from earlier observations and implications for future research of wider interest. I am still not fully satisfied with the raw fork data processing and believe that such a well-known group could have done a better job. I can recommend publication in Nature Communications with one additional clarification: In the caption of Figure 2 it should be clearly stated that the linear behavior of Q at higher temperatures extrapolating to the origin is not an independent measurement result, but a feature imposed in the data processing.

Reviewer #2 (Remarks to the Author):

The authors have addressed all comments raised in my previous report and sufficiently improved the manuscript. So, I would recommend this paper for publication in Nature Communications as it is.

Reviewer #3 (Remarks to the Author):

I can confirm my concerns have been expertly dealt with by the authors. It also seems to me that the concerns expressed by the other referees have been removed in the revised version, but I leave this conclusion to the other referees. I can thus support publishing this article in Nature Communications.

Response to reviewers comments

We thank all the reviewers for their reviews in the 2nd round.

In response to Reviewer #1

“The authors took into account all technical suggestions and also improved discussion of physics, to demonstrate clearer the difference of this work from earlier observations and implications for future research of wider interest. I am still not fully satisfied with the raw fork data processing and believe that such a well-known group could have done a better job. I can recommend publication in Nature Communications with one additional clarification: In the caption of Figure 2 it should be clearly stated that the linear behavior of Q at higher temperatures extrapolating to the origin is not an independent measurement result, but a feature imposed in the data processing.”

We have modified the Caption to figure 2. It now reads:

Figure 2: Quartz Fork Q vs Temperature a) The inferred Q of the quartz fork at various pressures vs the temperature. The expected Fermi liquid behavior was obtained after processing described in Supplementary Note 2, ($Q \propto \eta^{-1/2} \propto T$), and is represented by dashed lines. The superfluid transition is marked by an abrupt increase in the Q, and data below T_c is shown as lighter shaded lines. b) Plot of Q/T vs $(1 - T/T_c)$. This plot illustrates the extent of the departure of Q from linear behavior with pressure

We believe this is appropriate and is in accord with the reviewer’s intent.

Jeevak Parpia

For Authors